# Enhanced specificity of clinical high-sensitivity tumor mutation profiling in cell-free DNA via paired normal sequencing using MSK-ACCESS

A. Rose Brannon [1,6], Gowtham Jayakumaran[1,6], Monica Diosdado [1], Juber Patel[2], Anna Razumova[1], Yu Hu[1], Fanli Meng[2], Mohammad Haque[1], Justyna Sadowska[1], Brian J. Murphy[1], Tessara Baldi[1], Ian Johnson [2], Ryan Ptashkin[1], Maysun Hasan [2], Preethi Srinivasan[2], Anoop Balakrishnan Rema [1], Ivelise Rijo[1], Aaron Agarunov[1], Helen Won[2], Dilmi Perera[2], David N. Brown[2], Aliaksandra Samoila[3], Xiaohong Jing[2], Erika Gedvilaite [1], Julie L. Yang [2], Dennis P. Stephens[2], Jenna-Marie Dix [1], Nicole DeGroat[1], Khedoudja Nafa[1], Aijazuddin Syed[1], Alan Li[2], Emily S. Lebow[4], Anita S. Bowman [1], Donna C. Ferguson[1], Ying Liu[1], Douglas A. Mata[1], Rohit Sharma[1], Soo-Ryum Yang[1], Tejus Bale[1], Jamal K. Benhamida [1], Jason C. Chang[1], Snjezana Dogan[1], Meera R. Hameed[1], Jaclyn F. Hechtman[1], Christine Moung[1], Dara S. Ross[1], Efsevia Vakiani[1], Chad M. Vanderbilt [1], JinJuan Yao[1], Pedram Razavi [5], Lillian M. Smyth [5], Sarat Chandarlapaty [5], Gopa Iyer [5], Wassim Abida[5], James J. Harding[5], Benjamin Krantz[5], Eileen O'Reilly [5], Helena A. Yu[5], Bob T. Li [5], Charles M. Rudin [5], Luis Diaz [5], David B. Solit [2,5], Maria E. Arcila[1], Marc Ladanyi[1], Brian Loomis [2], Dana Tsui[1,2,7], Michael F. Berger [1,2,7], Ahmet Zehir [1,7✉] & Ryma Benayed [1,7✉]

Circulating cell-free DNA from blood plasma of cancer patients can be used to non-invasively interrogate somatic tumor alterations. Here we develop MSK-ACCESS (Memorial Sloan Kettering - Analysis of Circulating cfDNA to Examine Somatic Status), an NGS assay for detection of very low frequency somatic alterations in 129 genes. Analytical validation demonstrated 92% sensitivity in de-novo mutation calling down to 0.5% allele frequency and 99% for a priori mutation profiling. To evaluate the performance of MSK-ACCESS, we report results from 681 prospective blood samples that underwent clinical analysis to guide patient management. Somatic alterations are detected in 73% of the samples, 56% of which have clinically actionable alterations. The utilization of matched normal sequencing allows retention of somatic alterations while removing over 10,000 germline and clonal hematopoiesis variants. Our experience illustrates the importance of analyzing matched normal samples when interpreting cfDNA results and highlights the importance of cfDNA as a genomic profiling source for cancer patients.

[1] Department of Pathology, Memorial Sloan Kettering Cancer Center, New York, NY, USA. [2] Kravis Center for Molecular Oncology, Memorial Sloan Kettering Cancer Center, New York, NY, USA. [3] Department of Laboratory Medicine, Memorial Sloan Kettering Cancer Center, New York, NY, USA. [4] Department of Radiation Oncology, Memorial Sloan Kettering Cancer Center, New York, NY, USA. [5] Department of Medicine, Memorial Sloan Kettering Cancer Center, New York, NY, USA. [6]These authors contributed equally: A. Rose Brannon, Gowtham Jayakumaran. [7]These authors jointly supervised this work: Dana Tsui, Michael F. Berger, Ahmet Zehir, Ryma Benayed. ✉email: zehira@mskcc.org; benayedr@mskcc.org

Advances in molecular profiling have led to a rapid expansion in the number of predictive molecular bio-markers and associated targeted therapies, heightening the need for large-scale, prospective tumor profiling assays across all cancer types. The majority of comprehensive next-generation sequencing (NGS)-based profiling methods utilize tumor tissue as the primary specimen of choice for biomarker detection. Although widely used, obtaining an adequate tissue sample can be challenging in some cases due to the need for invasive biopsies that may pose an excessive risk to the patient. In addition, based on our clinical experience, 8.8% of the tissue submitted for molecular analysis is inadequate for testing due to low tumor cellularity, low DNA yield, or quality[1]. Finally, a single tissue biopsy may not capture the full genetic heterogeneity of a patient's cancer, and consequently, clinically actionable bio-markers may be overlooked even with the most sensitive and specific genomic assay. Taken together, a sole tissue-based genomic profiling approach may not be comprehensive and may limit treatment options for cancer patients.

The successful detection of cancer drivers in circulating-tumor DNA (ctDNA) found within plasma cell-free DNA (cfDNA)[2] has provided a means to overcome the limitations of tissue profiling[3,4]. cfDNA profiling can have a direct impact on patient care by informing treatment decisions[5,6], enabling the monitoring of cancer response to therapy[7,8], revealing drug resistance mechanisms[9,10], and detecting minimal residual disease or relapse[11–13]. In addition, by providing a less invasive collection procedure, cfDNA analyses also enable serial molecular profiling during the course of the patient's disease[14,15]. Plasma profiling can also potentially capture inter- and intra-tumor heterogeneity across disease sites especially in patients with advanced metastatic disease[16,17]. In addition, recent studies have shown that ctDNA fragmentation profiles can better facilitate cancer screening and early diagnosis[18].

The use of ctDNA as an analyte, however, has its inherent limitations. It is usually found in low concentrations in the plasma[19], which may be the result of low disease burden in early-stage tumors, disease control in response to treatment, or low tumor DNA shedding in blood. Moreover, the vast majority of cfDNA is typically derived from normal hematopoietic cells, leading to low levels of ctDNA and very low mutant allele frequencies for somatic mutations. Highly sensitive assays that are limited to single mutation ctDNA profiling assays such as droplet digital PCR (ddPCR)[20] are not practical for broad clinical use given the increasing number of genomic alterations that are predictive of response to FDA-approved targeted therapies or required as inclusion criteria for clinical trial enrollment. Given the low levels of ctDNA in a blood sample, the development of a highly sensitive NGS assay that comprehensively encompasses all clinically actionable targets is crucial for the detection of more low-frequency alterations. Advances in NGS technologies, such as the introduction of unique molecular identifiers (UMIs) and dual barcode indexing, have enabled ultra-deep sequencing of cfDNA while dramatically reducing background error rates, thereby allowing high-confidence mutation detection of very low allele frequencies[21]. Further, technical improvements in sequencing library preparation methods have reduced the input DNA required for sequencing, allowing for the efficient generation of libraries with input DNA as low as 10 ng.

Herein, we describe the design, analytical validation, and clinical implementation of MSK-ACCESS (Memorial Sloan Kettering - Analysis of Circulating cfDNA to Examine Somatic Status) as a clinical test that can detect all classes of somatic genetic alterations (single nucleotide variants (SNVs), indels, copy number alterations, and structural variants (SV)) in cfDNA specimens. This assay utilizes hybridization capture and deep sequencing (~20,000× raw coverage) to identify genomic alterations in selected regions of 129 key cancer-associated genes. MSK-ACCESS was approved for clinical use by the New York State Department of Health on 31 May 2019, and has since been used prospectively to guide patient care. We therefore also report our clinical experience utilizing MSK-ACCESS to prospectively profile 681 clinical blood samples from 617 patients, representing a total of 31 distinct tumor types.

## Results

**Panel design and background error assessment**. We utilized genomic data from over 25,000 solid tumors sequenced by MSK-IMPACT to generate a list of 826 exons from 129 genes encompassing the most recurrent oncogenic mutations; variants that are targets of approved or investigational therapies based on OncoKB, an in-house, institutional knowledge base of variant annotations[22]; frequently mutated exons; entire kinase domains of targetable receptor tyrosine kinases; and all coding exons of selected tumor suppressor genes. This MSK-IMPACT-informed design targets an average of three non-synonymous mutations and at least one non-synonymous mutation in 84% of the 25,000 tumors previously sequenced using MSK-IMPACT, including 91% of breast cancers and 94% of non-small cell lung cancers (NSCLC) (Fig. 1a). To further expand the detection capability of copy number alterations and SV in 10 genes, we additionally targeted 560 common SNPs and 40 introns known to be involved in rearrangements. MSK-ACCESS incorporates unique molecular indexes (UMIs) to increase fidelity of the sequencing reads. The overall process (Fig. 1b) involves the sequencing of plasma cfDNA and genomic DNA from white blood cells (WBCs) to ~20,000× and 1000× raw coverage, respectively, followed by collapsing read pairs to duplex (both strands of the initial cfDNA molecule) or simplex (one strand of the initial molecule) consensus sequences based on UMIs to suppress background sequencing errors. Duplex coverage represents the number of unique double-stranded cfDNA molecules for which both complementary strands were successfully sequenced. Replicate read pairs from both strands are jointly collapsed into a duplex consensus sequence to ensure highest-stringency mutation calling. Only mutations present on reads originating from both strands of the cfDNA template are retained and considered high-confidence calls[23] (Fig. 1c).

We first sought to characterize the error rate of MSK-ACCESS using a cohort of 47 plasma samples collected from healthy donors. The donor plasma samples were sequenced to a mean raw coverage of 18,818×. Post collapsing, the mean simplex and duplex coverage were 658× and 1103×, respectively (Fig. 1d, Supplementary Fig. 1). When considering only the sites with background error across all targeted sites (i.e., positions with non-reference alleles), we observed a median error rate of $1.2 \times 10^{-5}$ and $1.7 \times 10^{-6}$ in simplex and duplex BAM files, respectively, compared to a median of $3.3 \times 10^{-4}$ in the standard BAMs (Fig. 1e). Compared to the relatively equivalent background error rate on the HiSeq 2500 (Supplementary Fig. 2), the standard BAMs on the NovaSeq 6000 showed a higher error rate for T > A, T > G, and C > A base transversions (Fig. 1e). However, following collapsing, we observed lower and more uniform error rates across both sequencers. Moreover, while <1% of targeted positions in the standard BAM had an error rate of zero, 85% of positions in the simplex BAM and 99% in duplex BAMs had no observed base pair mismatches (Fig. 1f).

**Analytical validation**. For the analytical validation of MSK-ACCESS, we assembled a cohort of 70 cfDNA samples with a total of 100 known SNVs and indels in *AKT1, ALK, BRAF, EGFR, ERBB2, ESR1, KRAS, MET, PIK3CA,* and *TP53* identified by

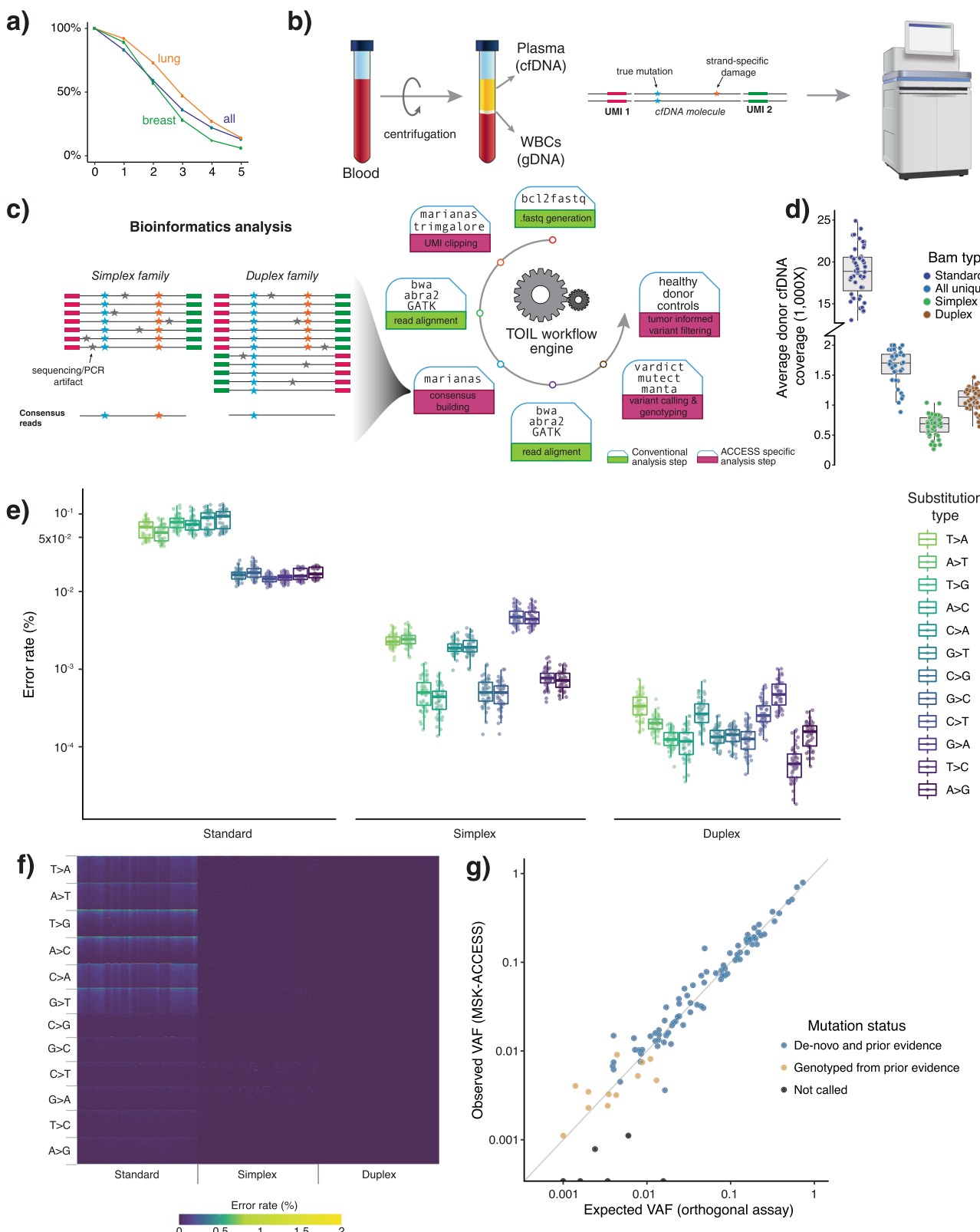

orthogonal cfDNA assays (ddPCR or a commercial NGS assay) from the same specimen to demonstrate accuracy. The range of VAF for the expected mutations, based on orthogonal assays, was 0.1–73%. We detected 94% of the expected variants ($n = 94$, 95% CI: 87.4–97.8%) based on genotyping and 82% of them ($n = 82$, 95% CI: 72.7–88.7%) with de novo mutation calling ($R^2 = 0.98$) (Fig. 1g, Supplementary Data 1, Supplementary Table 1).

Amongst the undetected mutations, leftover DNA was available for only one of the samples (orthogonal VAF = 0.16%), and ddPCR testing of this sample revealed no evidence of the alteration in our specimen. For mutations with VAF ≥ 0.5% from orthogonal assays ($n = 83$), we called 92% ($n = 76$, 95% CI: 84–96.5%) de novo, and we detected 99% of the mutations by genotyping ($n = 82$, 95% CI: 92.5–99.9%).

**Fig. 1 Design, characterization, and validation of MSK-ACCESS. a** The MSK-ACCESS panel was designed using data from 25,000 tumors analyzed using MSK-IMPACT tumor sequencing assay to identify at least one mutation in 94% of lung cancers, 91% of breast, and 84% of all cancers. **b** The laboratory workflow includes the extraction of cfDNA from plasma and genomic DNA from WBC originating from the same tube of blood. The addition of UMIs during library construction enables the identification of original cfDNA molecules during analysis and error suppression. **c** The analysis pipeline is modified from the standard MSK-IMPACT pipeline to incorporate UMI clipping and the generation of simplex and duplex consensus reads. **d** The sequencing of healthy donors to a mean raw coverage of 18,818× yielded a mean duplex coverage of 1103× and a mean simplex coverage of 658× across 47 samples. **e** The background error rate of non-reference sites demonstrates the reduction of overall and substitution specific errors via consensus read generation. Only the genomic position with non-reference reads are used; error rate is defined as the percentage of reads that support non-reference alleles. $N = 47$ for each boxplot. **f** A heatmap of error rate at all positions demonstrates how effective consensus read generation is at decreasing the error to zero at over 85% of sites. **g** Comparison of orthogonal and validated testing (expected VAF) to MSK-ACCESS (observed VAF) in the accuracy analysis showed high concordance ($R^2 = 0.98$). All boxplots show the median (center line) and 25th and 75th percentiles (bounding box) along with the 1.5 interquartile range (whiskers).

To determine the reproducibility of the assay, we prepared and sequenced seven samples, harboring a total of 152 mutations, both three different times within the same sequencing run and also across four separate runs (Supplementary Data 2). By genotyping, we detected 99% ($n = 151$, 95% CI: 96.4–100%) of the expected mutations with an overall median coefficient of variation of 0.16 (range: 0.04–1.2) for each sample and alteration. To test the limit of detection of the assay, we sequenced five different dilution levels (5, 2.5, 1, 0.5, and 0.1%) with a positive control sample containing 19 known mutations. In the 0.1% dilution, 11% of the mutations ($n = 2$, 95% CI: 1.3–33.1%) were called de novo and 74% ($n = 14$, 95% CI: 48.8–90.9%) were detected by genotyping. All expected mutations were called de novo in the 0.5% sample (Supplementary Data 3).

Finally, to calculate specificity, variant calling was performed on 47 healthy donor plasma samples in comparison to their matched WBCs, and no mutations were called. In addition, we utilized the samples from the accuracy analysis with orthogonal NGS results ($n = 37$), and considered all genomic positions interrogated by these assays ($n = 1620$) (Supplementary Data 4). Four potential false positives not reported by the orthogonal NGS assay (TP53 p.R253H with VAFs 0.17 and 0.24%, and PIK3CA p. H1047R with VAFs 0.05 and 0.07%) were detected by MSK-ACCESS using genotyping thresholds, implying a specificity of at least 99.7% (95% CI: 99.3–99.9%). Through de novo mutation calling, we identified only one false positive mutation, for a specificity of 99.9% (95% CI: 99.6–100%). Overall, our positive predictive agreement (PPA) for genotyping was 94% (95% CI: 85–98%) and for de novo mutation calling was 98% (95% CI: 90–99.9%). The negative predictive agreement (NPA) was 99.7% and 99.2% for genotyping and de novo calling, respectively.

**Clinical experience—genomic landscape**. Based on the above analytical validation results, MSK-ACCESS received approval for clinical use from the New York State Department of Health (NYS-DOH) on 31 May 2019 and was subsequently launched for routine clinical diagnostics assessment. Here, we describe the results from the first 617 patients prospectively sequenced in our clinical laboratory. A total of 687 blood samples were accessioned, and 681 (99%) yielded sufficient cfDNA and passed quality control metrics. The median raw coverage of the plasma isolated from these blood samples were 18,264× and 1273× for WBCs. Median duplex consensus coverage for plasma was 1497×.

Of the 681 samples, 51% ($n = 349$) were from NSCLC patients, followed by prostate, bladder, pancreatic, and biliary samples as the next most common cancer types (28%) (Fig. 2a). We assessed the clinical actionability of genomic alterations detected by MSK-ACCESS using OncoKB, and 41% ($n = 278$) of samples had at least one targetable alteration as defined by the presence of an OncoKB level 1-3B alteration. The highest frequency of level 1 OncoKB alterations were observed in bladder cancer, breast cancer, and NSCLC patients at 48%, 37%, and 33%, respectively.

Seventy-three percent ($n = 498$) of all samples had at least one alteration (mutation, copy number alteration or a SV) detected, with a non-zero median of 3 per patient (range 1–28) (Fig. 2b), 56% of which harbored clinically actionable alterations.

Altogether, we clinically reported a total of 1697 SNVs and indels in 486 samples from 435 patients, with a median VAF of 1.9% (range 0.02–99%) (Fig. 2c). Of these mutation calls, 95% ($n = 1606$) were called de novo without the aid of prior molecular profiling results for the tested patient. For the remaining 91 variants that were rescued by genotyping, the median observed VAF was 0.08%. As expected, deeper coverage enabled the detection of mutations at lower allele fractions for both de novo and genotyping thresholds (Fig. 2d). However, de novo calling of alterations that were independently seen previously in tumors occurred across the entire mathematically possible range, given minimum required alternate alleles, allele frequencies, and coverage depths (Fig. 2c, d).

To ensure the accurate identification of the expected alterations by our assay, we examined the most frequently called mutations, copy number alterations, and SVs in lung cancer and the next five largest disease cohorts (Fig. 2e). As expected, *TP53* was the most commonly altered gene, with variants in 144 of the 248 (58%) NSCLC samples with detectable alterations. Of greater therapeutic relevance, MSK-ACCESS identified oncogenic targetable driver mutations and amplifications in *EGFR, KRAS, MET, ERBB2,* and *BRAF*. Characteristically, lung cancer samples lacking known mitogenic drivers by MSK-ACCESS were found to harbor *STK11* and *KEAP1* mutations. *EML4-ALK* and *KIF5B-RET* fusions were also detected, de novo and by genotyping, in this cohort, along with rearrangements of *ROS1* with multiple partners.

Both clinically actionable and oncogenic alterations were similarly found in the next five most represented tumor types prospectively sequenced by MSK-ACCESS (Fig. 2e). *TP53* was again the most commonly altered gene, including both mutations and likely oncogenic deletions identified. *FGFR2* mutations and fusions (most commonly fused to *BICC1*) were identified in 8 of the 24 intrahepatic cholangiocarcinomas with detectable ctDNA, including missense mutations in the *FGFR2* kinase domain known to confer resistance to targeted therapies. Targetable alterations were also identified in *IDH1* and *PIK3CA*. Alterations in *FGFR3, ERBB2, AR,* and *KRAS* were recurrently detected in bladder, breast, prostate, and pancreatic cancer, respectively. Overall, the alteration rates in select genes and cancer types between MSK-ACCESS and MSK-IMPACT were comparable, with some notable exceptions such as a decrease in *KRAS* mutations in pancreatic cancer, an increase of *EGFR* mutations in the lung, *AR* mutations in prostate cancer, and *NF1* mutations in Breast cancer in the ctDNA (Fig. 2f).

**Concordance with MSK-IMPACT**. To compare the detection sensitivity and the spectrum of mutations observed between tumor tissue and plasma, we sought to examine the concordance

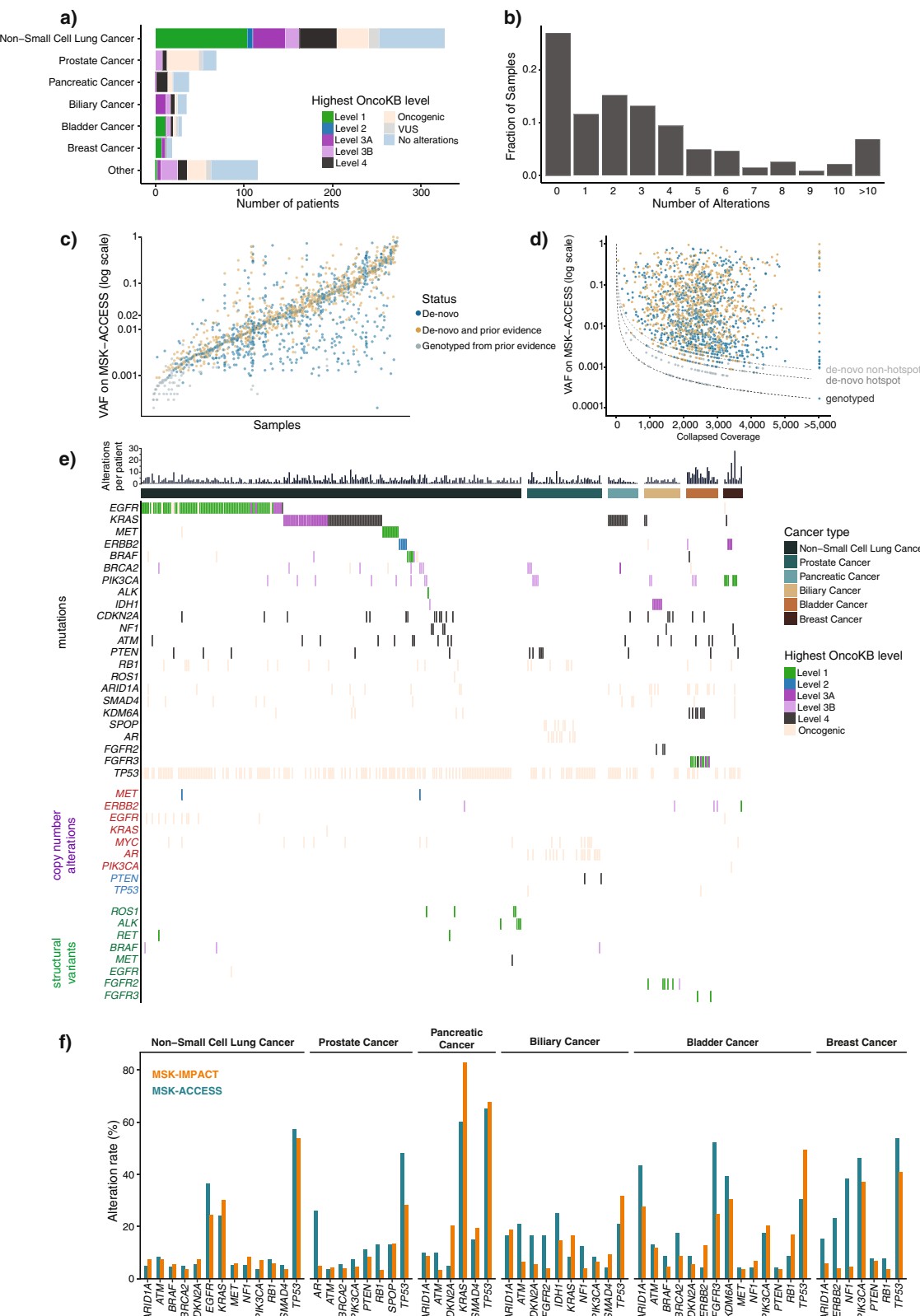

of mutation calling between MSK-IMPACT and MSK-ACCESS where available. For a consistent comparison analysis across all patients, we selected plasma mutations from the first sample sequenced by MSK-ACCESS for each patient for whom multiple time points were analyzed, and used the union of mutations across all tissue samples for each patient sequenced on MSK-IMPACT. Of the 617 patients tested with MSK-ACCESS, 383 also

had clinical MSK-IMPACT results from 520 sequenced tumor tissues. A total of 1206 mutations were reported in the overlapping target regions across both assays, and 59% ($n = 706$) of the mutations were reported by both assays (Fig. 3a). The distribution of allele frequencies in tissue was slightly higher for the shared mutations than for the MSK-IMPACT-only calls ($p = 2.06 \times 10^{-18}$), but this effect was not observed for the

**Fig. 2 Clinical experience with MSK-ACCESS. a** Distribution of cancer types amongst the first 617 patients sequenced with MSK-ACCESS. Colors indicate the highest OncoKB level ascribed to each patient's genomic findings. **b** Distribution of all alterations found in each ctDNA sample ($n = 681$). **c** Variant allele frequencies (VAF) of all mutations found in ctDNA samples from MSK-ACCESS. Samples were sorted by the median VAF and each mutation was colored based on whether prior evidence was found for the mutation. De novo: mutations were called in ctDNA and were not reported in tissue testing or tissue testing was not performed; De novo and prior evidence: mutations were called in ctDNA and also were present in tissue testing; Genotyped from prior evidence: mutations were not detected in ctDNA by genotyping based on tissue results. **d** Same mutations in *c* showing the distribution of total collapsed coverage and VAF. Dotted line indicates the theoretical limits of calling threshold. **e** Oncoprint of genomic alterations found in lung, biliary, bladder, breast, prostate and pancreatic cancer samples with reported alterations. Colors indicate the OncoKB levels as in (**a**). **f** Comparison of cohort alteration rate of tumor types in (**e**) for genes where the alteration rate was greater than 3% by both MSK-ACCESS and MSK-IMPACT.

MSK-ACCESS-only calls (Fig. 3b). While the VAFs of shared mutations in tissue and plasma were weakly correlated, we nonetheless observed high-frequency tumor mutations at extremely low VAF by MSK-ACCESS, and vice versa (Fig. 3c).

We next considered the alterations specific to one assay. Twenty-one and twenty percent of the mutations were reported individually by either MSK-IMPACT tumor sequencing ($n = 254$) or MSK-ACCESS plasma sequencing ($n = 246$), respectively (Fig. 3a). Interestingly, 58 of 246 mutations reported by MSK-ACCESS-only were present at low sub-threshold levels in tissue by MSK-IMPACT, highlighting the potential for increased sensitivity obtained by utilizing ultra-high depth of coverage and UMIs. Eighteen percent ($n = 46$) of the MSK-IMPACT-only mutations were clinically actionable (OncoKB Level 1-3), as were 12% ($n = 30$) of the MSK-ACCESS-only detected mutations (Supplementary Fig. 3), clearly demonstrating the importance and value of complementary tissue and cfDNA analyses. Moreover, for patients that did not receive MSK-IMPACT testing ($n = 234$), MSK-ACCESS detected 79 total clinically actionable mutations in 26% ($n = 61$) of the patients.

In order to evaluate whether the tumor content played a role in mutation discordance, we computationally estimated tumor purity of tissue samples ($n = 433$ samples from 331 of the 383 patients for whom purity can be determined by FACETS, see "Methods") and found no significant difference between samples from patients who had mutations detected only in plasma vs both plasma and tissue ($p = 0.812$) (Fig. 3d). This was also true when purity was compared for patients with only actionable mutations (Supplementary Fig. 4). Next, we evaluated the clonality of mutations (see "Methods") in tissue samples of patients tested on both MSK-IMPACT and MSK-ACCESS. A larger proportion of shared mutations were clonal compared to tissue-only mutations ($p = 8.10 \times 10^{-13}$), which was also true when only actionable mutations were considered ($p = 6.06 \times 10^{-3}$) (Fig. 3e), indicating the role clonality plays in mutation concordance between tissue and plasma testing.

We also assessed whether the time between the tissue collection for MSK-IMPACT and blood collection for MSK-ACCESS (difference in date of procedure, ΔDOP) had an impact on the mutation concordance (see "Methods"). Overall, the average ΔDOP was 65 weeks (median: 27 weeks, range: 0–679 weeks). Patients with mutations only detected on either MSK-IMPACT (mean = 598 days; median = 491 days, $p = 3.63 \times 10^{-14}$) or MSK-ACCESS (mean = 616 days; median = 288 days, $p = 1.33 \times 10^{-9}$) showed a higher ΔDOP than patients for whom mutations were detected on both assays (mean = 351 days; median = 83 days; Fig. 3f, Supplementary Table 2). Evaluation of ΔDOP based only on actionable mutations across the three categories—MSK-IMPACT only, shared mutations, and MSK-ACCESS only—yielded similar findings (Supplementary Fig. 5). Taken together, these results underlie the importance of timing of sample collection and tumor heterogeneity with respect to mutation concordance between tissue and plasma-based testing.

**Utility of matched WBC analysis.** Similar to MSK-IMPACT, MSK-ACCESS utilizes matched WBC sequencing to confidently identify and remove germline variants from cfDNA results. To quantify the benefit of matched WBC sequencing, we performed plasma-only variant calling in all clinical cases, resulting in 24,561 variant calls. We then simulated filtering criteria for unmatched sequencing, removing 14,508 variant calls (median: $14 \pm 8$ variants per sample), based on their presence in our curated plasma normal samples or in at least 0.5% of the population by gnomAD (Fig. 4a, Supplementary Fig. 6). We could further filter out 721 (7.2%) likely germline variants based on their VAF within the heterozygous germline variant VAF range (between 35 and 65% in both WBCs and cfDNA). However, using this VAF-based filtering would improperly remove a total of 70 verified somatic mutations from the cfDNA callset, 15 of which were clinically actionable (Supplementary Table 3). Therefore, 10,053 variants with a mean VAF of 4.7% (median: 0.05%) still remained after database-driven filtering, highlighting the utility of patient-matched WBC profiling to filter out definitive germline mutations.

Notably, we were able to use the sequenced WBC sample to correctly classify several events as germline that were included as somatic events by commercial providers. As an example, a commercial cfDNA test reported an *ATM* p.E522fs*43 mutations as somatic and suggested therapies for this alteration, but our matched analysis revealed the indel to be present at ~50% in both the plasma and WBC and clearly ruled it out as a germline event. We have similarly been able to reassign mutations in *TP53*, *BRCA2*, and *ROS1* that had been previously reported as somatic as germline variants. In addition, the use of WBC sequencing revealed the germline origin of observed copy number deletions in *ATM*, *BRCA2*, and for two patients with retinoblastoma, *RB1*, based on deletions in their matched WBC sample (Supplementary Fig. 7).

As previous reports have demonstrated that tumor- and normal-derived cfDNA may be distinguishable from genomic DNA by fragment length[24,25], we sought to confirm this observation in MSK-ACCESS data and use this information to better inform the origin of variants detected in cfDNA. The general fragment length distribution exhibited the expected bimodal cfDNA peaks around 161 and 317 base pairs, when factoring the trimming of 3 bases from read ends by the pipeline[26] (Fig. 4b). For all cfDNA fragments harboring a somatic tumor-derived mutation confirmed to be absent in WBCs ($n = 1558$), we observed that these fragments were significantly shorter than those harboring the wild-type allele, consistent with their tumor origin (Fig. 4c–i) (bootstrapped $p$ value < 0.0001). In several variants with limited supporting evidence in WBC DNA but significantly greater VAF in plasma cfDNA we were able to distinguish the origin as somatic tumor-derived (ctDNA) nature based on the slight cfDNA insert size profile peak in the WBC sample. As demonstrated in Fig. 4c-II, these reads did have a shortened fragment length (bootstrapped $p$ value < 0.0001), confirming that they originated from the cell-free compartment.

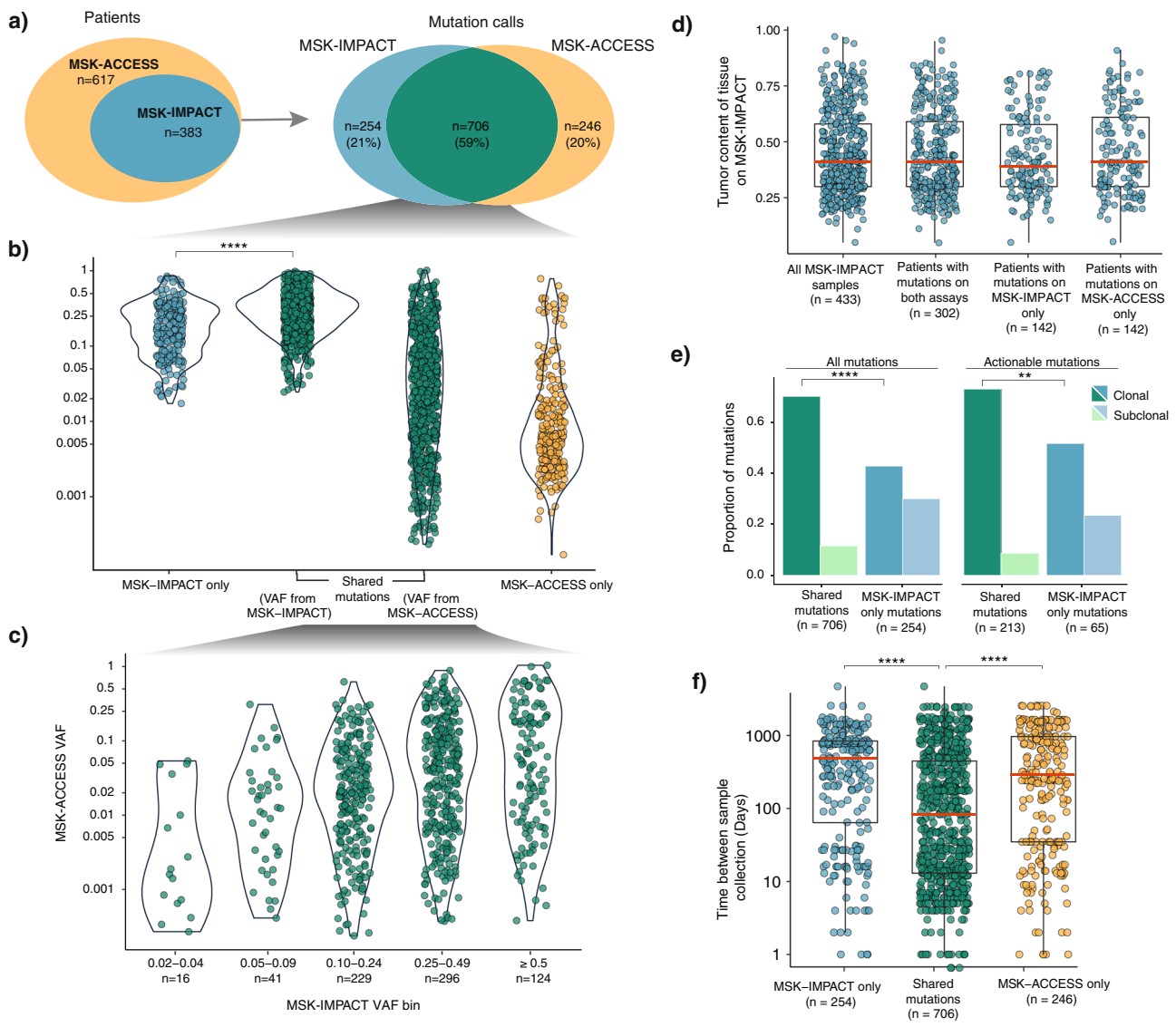

**Fig. 3 Comparison of mutation calls between ctDNA and tissue.** Comparison of mutation calls between ctDNA and tissue. **a** Venn diagrams indicating the number of samples with concurrent cfDNA and tissue testing ($n = 383$) and the number of mutation calls identified in each (total $n = 1206$). **b** VAF distribution of mutations identified by MSK-ACCESS-only, shared by both MSK-ACCESS and MSK-IMPACT, and by MSK-IMPACT only ($p = 2.06 \times 10^{-18}$). The p value was obtained from pairwise comparisons using two-sided Mann–Whitney U-tests and adjusted for multiple testing using the Bonferroni method. **c** Comparison of VAF distributions of mutations identified in both the ctDNA and tissue from both MSK-ACCESS and MSK-IMPACT. **d** Tumor purity distribution of MSK-IMPACT tissue samples of (i) all patients in the concordance analysis, (ii) samples belonging to patients presenting actionable mutations on both assays, (iii) samples belonging to patients with actionable mutations detected in MSK-IMPACT only, and (iv) samples belonging to patients with actionable mutations detected in MSK-ACCESS only. **e** Clonality of all and actionable mutations detected in MSK-IMPACT only and in both assays (all mutations: $p = 8.10 \times 10^{-13}$, actionable mutations: $p = 6.06 \times 10^{-3}$). The p values were obtained from two-by-two Fisher's exact tests and adjusted for multiple testing using the Bonferroni method. **f** Absolute time difference (ΔDOP) between MSK-IMPACT tissue sample and MSK-ACCESS blood sample collection for patients with actionable mutations in MSK-IMPACT only ($p = 3.63 \times 10^{-14}$), in both assays, and in MSK-ACCESS only ($p = 1.33 \times 10^{-9}$). The p values were obtained from pairwise comparisons using two-sided Mann–Whitney U-tests and adjusted for multiple testing using the Bonferroni method. All boxplots show the median (center line) and 25th and 75th percentiles (bounding box) along with the 1.5 interquartile range (whiskers).

In stark contrast, the variant calls from the unmatched analysis that were filtered out as putative germline variants by their presence in WBCs at high VAF demonstrated an equivalent fragment length distribution as wild-type alleles (Fig. 4c-III) (bootstrapped p value = 0.94). As we have shown, by integrating the fragment length analysis into the MSK-ACCESS assay, we can confidently distinguish between tumor-derived somatic and normal-derived variants in cfDNA.

**Assessing the filtering of putative clonal hematopoiesis (CH) mutations.** Several recent studies have suggested that CH

mutations present a challenge for proper filtering in highly sensitive NGS-based liquid biopsy assays[27–30]. We observed that the use of patient-matched normal WBC DNA in MSK-ACCESS eliminated 7,760 (77%) of variant calls below 10% VAF (Fig. 4a-IV). We posited that the majority of these calls represent potential CH mutations. Recent reports[31] have suggested that fragments supporting CH variants have length distributions similar to cfDNA derived from non-cancerous cells and distinct from ctDNA[27,29,32]. Indeed, the sequence reads harboring variants with plasma VAF < 10% and present in WBCs exhibited fragment lengths indistinguishable from wild-type and germline variants

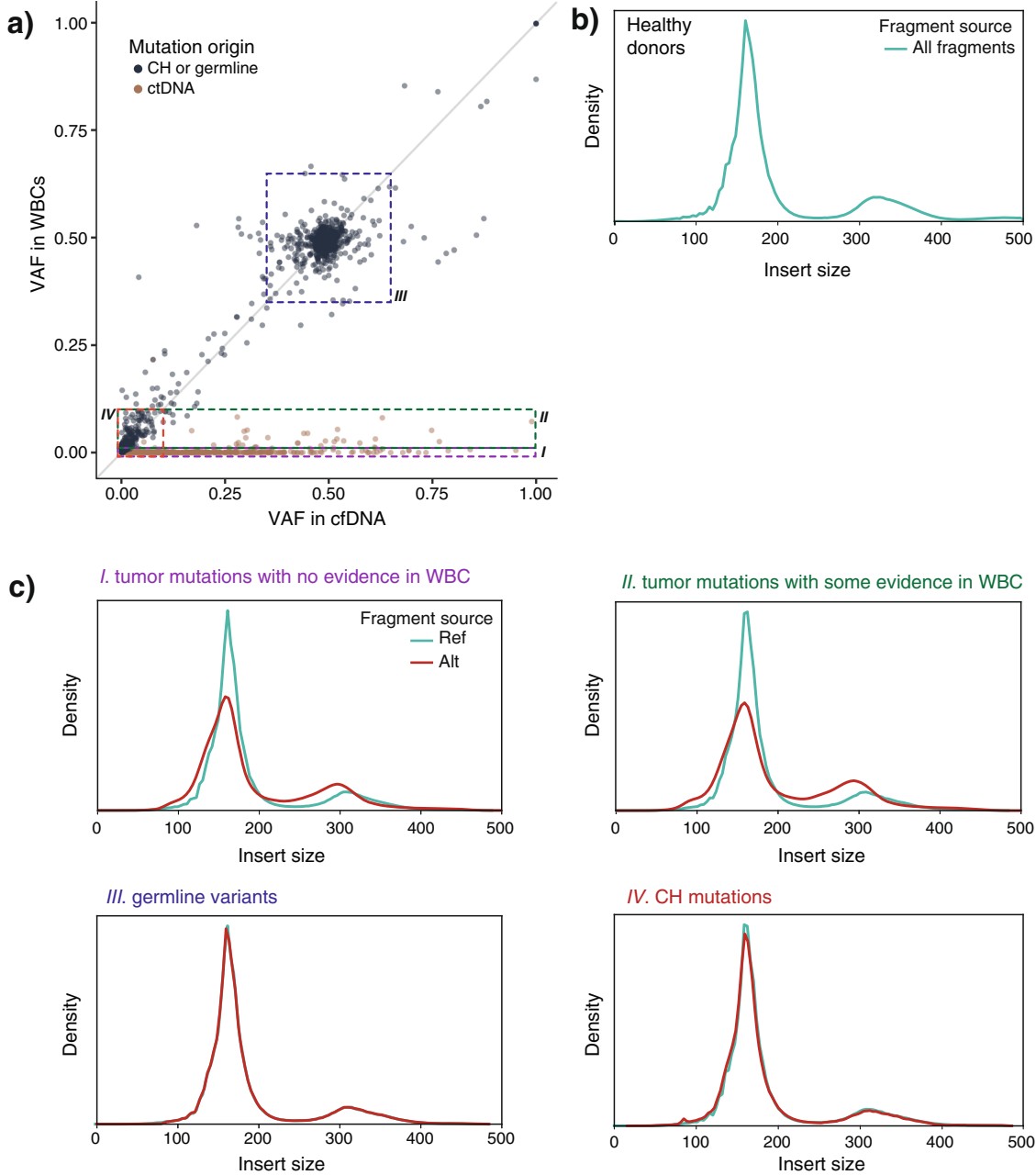

**Fig. 4 Use of WBC sequencing data to classify variants found in cfDNA. a** VAF distribution of all mutations called in plasma from cfDNA and WBCs. Colors indicate the origin of mutations. Boxes indicate different populations of mutations: I: Variants only present in cfDNA, II: Variants present in cfDNA at high VAF but also present in WBC at lower VAF, III: Variants present in both cfDNA and WBCs with VAFs in the presumed germline range (35–65%), IV: Variants present in both cfDNA and WBCs with VAFs lower than 10% in both. **b** Insert size distribution of sequencing reads (fragment size) in healthy donors with characteristic peaks at 161 bp and 317 bp **c** Fragment size distribution for reads encompassing the variants highlighted by the boxes and labels in (**a**) for both reference and alternate alleles. Clear differences are observed for reads originating from ctDNA vs normal tissue.

(Fig. 4c-IV) (bootstrapped *p* value = 0.99), adding confidence to the hypothesis that these were properly filtered WBC-derived somatic mutations associated with CH. The previously described alterations in Fig. 4c-II with a lower frequency of reads in the WBC sample than in the cfDNA sample could also have been interpreted as having a CH origin. Nonetheless, the shorter length distribution for fragments harboring these mutations reaffirmed that these were likely tumor-derived as originally postulated.

Given our ability to recognize CH from WBCs, we have been able to reclassify several variant calls reported as somatic events by commercial vendors. While some of these calls were in commonly mutated CH genes such as *DNMT3A*, some were in

less common genes. In one case, a patient with lung adenocarcinoma with an external report of *KRAS* p.G12S. However, we identified this alteration at equivalent frequencies (0.44 and 0.31%) in the plasma and WBC, suggesting that it most likely represents a CH mutation, underlying the complexities of assigning such alterations to different compartments when considering the clinical presentation of the patient.

The identification of driver genetic alterations in key oncogenes and tumor suppressor genes plays an essential role in the diagnosis and treatment of many cancers. For more than a decade, biomarker analyses have been predominantly accomplished in solid tumors by sequencing tumor tissue collected at

the time of surgical resection, diagnostic tumor biopsy, or cytology. However, in recent years, several studies have demonstrated that "liquid biopsies" could provide similar, and in some cases, more comprehensive information accompanied by a less invasive approach. Due to the significantly reduced procedure risk, they also enable longitudinal monitoring, which can substantially impact patient management. Here, we describe the analytical validation and clinical implementation of MSK-ACCESS, a hybridization capture-based NGS assay comprising 129 genes and capturing multiple classes of genomic alterations (SNV, indels, copy number alterations, and SV). Because of the scarcity of cfDNA material in the plasma and even smaller amounts of ctDNA, the development of this assay was guided by two key considerations: First, the assay had to enable the detection of low-frequency genomic alterations; and second, it had to incorporate the implementation of matched WBC sequencing to effectively filter out germline variants and CH mutations.

Our analytical validation of MSK-ACCESS was performed using 70 cfDNA samples known to be positive for mutation hotspots using orthogonal methods. MSK-ACCESS has demonstrated a low background error rate, a 92% de novo sensitivity down to 0.5% VAF for SNVs and indels, and a 99% specificity. Following approval by the NYS Department of Health, we prospectively sequenced 681 plasma samples from 617 unique patients with non-small lung cancer, prostate, bladder, pancreatic, and biliary cancer most commonly. Alterations were detected in 73% of all prospective clinical samples, with some of the negative cases representing patients with known disease control or in the post-operative setting. Clinically actionable alterations were called in 41% of all samples and 56% of samples with alterations. Mutations were detected with VAF as low as 0.02%. While, we did leverage available MSK-IMPACT data to genotype prior mutations for higher sensitivity at lower allele frequencies, 95% of mutations were called de novo without the need for additional data.

In our clinical cohort, 62% of patients had a patient-matched tissue specimen analyzed using MSK-IMPACT. A total of 260 mutations found in the tissue were not reported by MSK-ACCESS. This discordance could be the result of a very low tumor fraction in cfDNA, tumor heterogeneity, or differential shedding into the plasma by different tumor sites. Therefore, we do not believe that plasma cfDNA profiling can replace tissue testing in all situations. Studies are ongoing to better elucidate the clinical and analytic factors that may lead to a lack of mutation detection in the cfDNA of such discordant cases. In addition, 250 mutations were detected by MSK-ACCESS but not reported in the patient tissue, and 12% of those were actionable. These MSK-ACCESS specific alterations are likely due in part to the inherent spatial/temporal limitations of tissue profiling, though in some cases it represented acquired drug resistance, such as the identification of FGFR3 point mutations known to confer resistance to FGFR inhibitor therapy in the FGFR3-TACC3 bladder cancers. As comprehensive data on oncologic therapy becomes available, the detailed mechanisms of discordance between tissue and plasma-derived mutations can be investigated further in future studies.

Taken together, our clinical experience has shown the importance of deep sequencing and the inclusion of matched WBCs to achieve high sensitivity and specificity to detect mutations in cfDNA. In addition, it has also demonstrated that tissue and cfDNA based sequencing approaches are complementary in certain cases and can be used to effectively and comprehensively detect all classes of genomic alterations. Specifically, 91 out of the 1697 mutations detected by MSK-ACCESS with a median VAF of 0.08% were rescued by genotyping based on events called previously by MSK-IMPACT

in the matched tissue. Conversely, 26% of patients in this cohort harbored at least one actionable mutation not previously known from tumor tissue profiling.

As the ability to detect ctDNA in the minimal residual disease setting is proportional to the number of mutations interrogated, a uniform panel such as MSK-ACCESS will exhibit differential sensitivity across patients with variable mutation burden. In most cases, MSK-ACCESS will not be as sensitive as patient-specific bespoke panels customized to detect dozens or more mutations identified from a tumor exome or genome[33]. However, the path to clinical validation and operationalization of a patient-specific approach is uncertain and unattainable to most laboratories. Moreover, the design of MSK-ACCESS incorporates the most frequently mutated genomic regions from a cohort of more than 25,000 solid tumors clinically profiled, thereby maximizing the number of mutations that may be genotyped and monitored throughout treatment for a standardized assay.

Correctly classifying mutations associated with CH represents a major challenge for all blood-based liquid biopsy assays. Commercially reported CH mutations could be misconstrued as recurrence when in fact no recurrence may be present or wrongfully considered as a tumor mutation and tracked across multiple blood draws for monitoring of response to therapy. By incorporating the sequencing of a time-matched WBC sample, we significantly decrease the likelihood of calling and reporting CH alterations that are frequently observed in commercial tests that do not include the normal DNA. CH[27,34] mutations that occur at very low allele frequencies may be incorrectly classified as tumor-derived somatic mutations when they are detected in individual cfDNA molecules but not in WBC DNA. Reassuringly, our analysis suggests that our approach effectively eliminates a large number of CH events that exhibit fragment length characteristics consistent with hematopoietic cell-derived rather than tumor cell-derived cfDNA.

In conclusion, MSK-ACCESS can be used to detect clinically relevant alterations through a less invasive mechanism than tumor biopsies, better enabling treatment decisions. The use of WBCs from the same blood draw limits the improper reporting of germline and CH alterations, which allows for more accurate reporting of somatic alterations. By automatically integrating prior patient-specific results into the analysis of plasma sequencing data, liquid biopsy profiling can provide a more sensitive and comprehensive representation of the genomic makeup of a patient's cancer, enabling improved patient care.

## Methods

**Cohort information**. All samples were collected with informed consent from the patients for routine prospective clinical genomic analyses. Clinical sequencing data from 681 patients who were enrolled in an IRB-approved research protocol (MSKCC; NCT01775072) were used. This study was approved by the MSKCC Institutional Review Board/Privacy Board.

**Panel design**. Probes (120 bp long) were designed to cover the entire length of 826 exons and 40 introns of 129 genes, targeting ~400 kilobases of the human genome. Probes were divided into 2 pools: Pool A included regions covering protein-coding exons for the detection of SNVs and indels, as well as 171 microsatellite regions for the detection of microsatellite instability (MSI). Pool B included regions covering introns for the detection of gene fusion breakpoints in 10 genes and SNPs for quality control and improved detection of copy number alterations. Pool A and Pool B were combined in a 50:1 ratio to efficiently distribute sequence reads such that, in a single capture reaction, we achieved ultra-deep raw sequencing coverage (12,000–25,000×) for Pool A targets and standard raw sequencing coverage (500–1500×) for Pool B targets. For matched WBC samples, a 1:1 ratio of Pool A to Pool B was used.

**cfDNA extraction, library construction, and capture**. For each patient, with appropriate informed consent, both cfDNA and WBC DNA were extracted from plasma (MagMAX cfDNA isolation kit) and buffy coat (Chemagen magnetic bead technology). Whenever possible, 20 ng of plasma cfDNA was used, but analyses

were attempted for samples with as little as 3 ng plasma cfDNA. UMIs and xGen Duplex Seq Adapters with dual index barcodes from IDT (Integrated DNA Technologies) were introduced during library construction. Libraries were pooled in equimolar concentrations and captured using the above-described custom IDT xGen Lockdown probes. Captured DNA fragments were then sequenced on an Illumina sequencer (HiSeq 2500 or NovaSeq 6000) as paired-end reads as described above for plasma cfDNA samples and to a target depth of ~1500× of raw sequencing coverage for WBCs.

**Analysis pipeline.** Sequencing data were demultiplexed with BCL2FASTQv2.1.9 (Illumina), UMIs were trimmed with Trim Galore (v0.2.5) and Marianas (https://github.com/mskcc/Marianas), and read pairs underwent alignment to the human GRCh37 reference genome with further post-processing using BWA MEM (v0.7.5a), ABRA2 (v2.17), and GATK (v3.3) to generate a "standard" BAM file. All pipeline workflows were built using the common workflow language (CWL) specification (https://www.commonwl.org/) and toil workflow engine[35]. Aligned PCR duplicates were collapsed into error-suppressed consensus reads based on UMI and position by Marianas. An additional three bases were trimmed from the ends of the collapsed reads due to increased sequencing errors at these positions. Collapsed and trimmed, reads were then re-aligned using the above standard pipeline. Collapsed BAM files include the "duplex" BAM with consensus reads generated from both strands of the original cfDNA template molecule, the "simplex" BAM with consensus reads generated from at least 3 reads of only one strand of the original template molecule, and the "all unique" BAM representing all sequenced template molecules: duplex consensus reads, simplex consensus reads, as well as sub-simplex consensus reads and singleton reads from 2 or 1 reads of one strand.

Variant calling was performed in a matched tumor-informed manner ("genotyping") using GetBaseCountsMultiSample (GBCM v.1.2.2, https://github.com/mskcc/GetBaseCountsMultiSample) when prior molecular profiling results were available for an individual. This genotyping method required at least 1 duplex or 2 simplex consensus reads, comprised of both Read1 and Read2, to call a SNV or indel at a site known to be mutated in a previous sample from that patient. De novo mutation calling by VarDict (v1.5.1) or MuTect (v1.1.5) required a minimum of 3 duplex consensus reads for a known cancer hotspot mutation or 5 for a non-hotspot mutation. Unless otherwise noted, reported variant allele depths (AD), total depth (DP), and allele frequencies (VAFs) represent the combined counts from simplex and duplex consensus reads. Copy number alterations were identified from the "all unique" BAM using a described previously method[22]. SV were called in the "standard" BAM files using Manta (v1.5.0)[23] and required a minimum of 3 fusion-spanning reads for a de novo SV or 1 fusion-spanning read for an SV previously identified in that patient.

Quality control metrics were calculated for all samples. Coverage and background error rate were calculated using Waltz (https://github.com/mskcc/Waltz). Base quality metrics per cycle were collected by Picard (v2.8.1). Plasma—normal matches were confirmed using a set of fingerprint SNPs.

Clinical actionability and treatment implications of specific cancer gene alterations were annotated using OncoKB (v2.2.0)[22].

**Error rate analysis.** Background error rate was characterized using Waltz, for pool A regions covering protein-coding exons for the detection of SNVs and indels (~200 kilobases). Using a cohort of 47 plasma samples from healthy donors, the error rate was calculated as fraction of reads supporting each of the substitution types across all targeted sites. A maximum allele frequency cut off of 2% was used in the error rate analysis. For each of the standard, duplex, and simplex bam types, median error rates were determined using the error rates for all substitution types across all 47 plasma samples. To determine the percentage of targeted sites with zero error rate, first, a matrix of error rate for all possible substitution types in each of the targeted sites across the 47 plasma samples was generated for standard, simplex, and duplex bams. For each substitution type at a given genomic position, the error rate was determined as the 95th percentile of the error rates across the 47 plasma samples. The substitution type error rates were then summarized to yield error rates for each of the targeted sites. Percentage of targeted sites with zero error rates were determined and reported for standard, simplex, and duplex bam types.

**Fragment size analysis.** Fragment size calculations were performed using the pysam module (https://github.com/pysam-developers/pysam). Read pairs, identified using SAM flags, mapping to all ACCESS targets were used to determine sample level fragment size distribution. Read pairs overlapping mutated loci that support either reference allele or variant allele were used to determine the size distribution of reference DNA fragments and mutated fragments, respectively. Analysis was restricted to fragments of size 500 bp or lower, which accounted for at least 95% of all fragments in a duplex plasma bam (Supplementary Fig. 8). Mutations with allele frequency lower than 0.05% in plasma samples were also excluded. Non-parametric bootstrap hypothesis testing was used to test the null hypothesis that the mean fragment sizes of reference and variant alleles are the same.

$$H_0 : \mu_{REF} = \mu_{ALT}$$

$$\Delta = \mu_{REF} - \mu_{ALT}$$

where, $\mu_{REF}$, is the mean fragment size of reference allele fragments, $\mu_{ALT}$, is the mean size of variant allele fragments, and, $\Delta$, is the test statistic. The null hypothesis was modeled using the data and the test statistic was calculated for 10,000 null datasets simulated using bootstrapping. Bootstrapped $p$ values were estimated based on the fraction of the time that the simulated dataset gave a statistic equal to or greater than the observed statistic in the original dataset.

**Performance statistic calculations.** Error assessment was calculated using plasma and matched WBCs collected from a cohort of 47 healthy donors (median age 29, range 21–48). The mean duplex consensus coverage for the normal plasmas was 1103X (sd = 181X). The error rate was calculated by averaging across all targeted genomic positions at non-SNP and in non-repetitive regions where the non-reference variant frequency was <2%.

The reference set for the analytical validation was generated from time-matched plasma samples or its extracted cfDNA tested with a validated ddPCR test or a commercial NGS assay. Seventy unique cfDNA samples from patient plasma, as well as SeraCare and AccuRef control samples, were used for assay validation. All calculations were performed for both genotyping of known variants as well as de novo calling thresholds. Sensitivity was calculated on patient samples as true positive (TP, called by both MSK-ACCESS and the orthogonal test) divided by all calls made orthogonally.

To calculate specificity, PPA, and NPA, we selected the 36 samples orthogonally sequenced by a commercial NGS assay and considered the 45 sites that had been called positive by at least one sample orthogonally, yielding 1620 total sites. A true negative (TN) was one called by neither MSK-ACCESS nor the orthogonal assay, a false positive (FP) was one called positive by MSK-ACCESS but negative by the orthogonal assay, and a false negative (FN) as negative by MSK-ACCESS but called orthogonally. Specificity was calculated as TN/(TN + FP). PPA was calculated as TP/(TP + FP), and NPA was calculated as TN/(TN + FN). For each, we also calculated Pearson exact confidence intervals at 95% power.

To calculate precision and reproducibility, the cfDNA from four patient samples, AccuRef 1% control, SeraCare 1%, and SeraCare 2.5% control samples were tested in triplicate within the same library preparation and sequencing run and in triplicate across multiple days of library construction and sequencing runs, each with a different barcode. To assess the limit of detection of the assay, 19 known mutations in SeraCare control samples at 5, 2.5, 1, 0.5, and 0.1% allele frequencies, and wild type were used.

**Clinical experience and concordance analysis.** For MSK-ACCESS/MSK-IMPACT concordance analysis, only the clinically reported mutation calls from the first MSK-ACCESS sample per patient were used when multiple samples were sequenced. The union of clinically reported mutation calls from all samples sequenced on MSK-IMPACT for each patient was used given that MSK-ACCESS could potentially overcome tumor heterogeneity. This combined mutation list was re-genotyped as fragments, or overlapping reads, using GBCM, and the genotyped values were used for downstream analyses. Where mutations were reported in multiple IMPACT samples, the maximum genotyped VAF was used. Calls labeled as sub-threshold had at least two supporting fragments.

For comparisons of the prevalence of alterations across patients, we used mutation data from 47,116 solid tumor samples sequenced with MSK-IMPACT and considered only mutations that intersected with the MSK-ACCESS target exons. Comparisons were performed for select genes and cancer types where the alteration rate was greater than 3% by both MSK-ACCESS and MSK-IMPACT.

Tumor content in tissues and clonality of mutations on MSK-IMPACT were estimated using FACETS[36] (v0.5.14, https://github.com/mskcc/facets). A mutation was classified as clonal if the cancer cell fraction value obtained from FACETS was greater than 80%, otherwise the mutation was classified as sub-clonal. Since a union of clinically reported mutation calls from all MSK-IMPACT samples per patient was used for the concordance analysis, tumor content of all MSK-IMPACT samples for these patients were used to assess the impact of tumor purity on mutation concordance. Mutations for which purity and clonality were indeterminate by FACETS, were excluded from analyses evaluating the impact of purity and clonality, respectively, on mutation concordance between MSK-IMPACT and MSK-ACCESS.

To assess the impact of time between sample collection for MSK-IMPACT and MSK-ACCESS on mutation concordance, we determined for each patient the absolute time difference between first blood collection for MSK-ACCESS and the tissue collection for MSK-IMPACT (ΔDOP) corresponding to a particular mutation.

**Reporting summary.** Further information on research design is available in the Nature Research Reporting Summary linked to this article.

## Data availability

The raw sequencing data are protected and are not available due to privacy laws. All results derived from the analysis of clinical sequencing data (mutations, copy number alterations, and structural variants) for all samples from 617 patients, including MSK-ACCESS and MSK-IMPACT, where applicable, are available through publicly accessible cBio Portal study (https://www.cbioportal.org/study/summary?id=msk_access_2021). The remaining data are available within the Article or Supplementary Information.

## Code availability

Analysis code for in-house developed pipeline modules are made available on Github. Marianas: https://github.com/mskcc/Marianas. Waltz: https://github.com/mskcc/Waltz. GetBaseCountsMultiSample: https://github.com/mskcc/GetBaseCountsMultiSample.

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

## Acknowledgements

We gratefully acknowledge Jesse Galle, Dalicia Reales, Risha Huq, Laetitia Borsu, Georgi Lukose, Ellinor Peerschke, William Leach, Mariam Khan, Sandy Naupari, Jake Bakas, Nelio Chaves, Shadia Islam, Yingjuan Xu, Hina Patel, Kizzia Carmel Perez, Srushti Kakadiya, Paulo Salazar, Brian Tedino, Elise Gallagher, Agnes Viale, and Kety H. Huberman for their important contributions. The study was supported by P30 core grant, Cycle for Survival, Marie-Josée and Henry R. Kravis Center for Molecular Oncology, NIH P50-CA221745, NIH P01-CA228696, R01-CA234361, the Wien Initiative in Liver Cancer Research, and the Society of MSKCC.

## Author contributions

M.F.B. and D.T. designed the panel and the assay. M.F.B., D.T., B.H.L., J.P., F.M., I.J., M.H., P.S., H.W., D.P., R.P., G.J., D.B., A.Z., E.G., X.J., and A.L. developed the assay. I.J., G.J., J.P., M.H., Y.H., and R.P., developed the bioinformatics pipeline. D.T., B.L., C.R., A.S., N.D., P.R., L.M.S., S.C., G.I., W.A., J.J.H., B.K., E.O., H.A.Y., and L.D. collected specimens for assay validation. R.B., A.Z., M.D., A.R.B., G.J., A.Z., B.H.L., J.P., M.F.B., D.T., Y.H., I.J., M.H., P.S., R.P., J.Y., D.S., J.S., B.J.M., J.D., N.D., and K.N. generated and interpreted the validation data. R.B., A.Z., M.D., A.R.B., G.J., A.R., M.H., J.S., B.J.M., T.B., R.P., A.S., A.B., A.A., E.L., K.N., M.E.A., M.L., A.S.B., D.C.F., Y.L., D.A.M., R.S., S.R.Y., T.B., J.K.B., J.C.C., S.D., M.R.H., J.F.H., C.M., D.S.R., E.V., C.M.V., J.J.Y., and I.R. generated and interpreted the clinical data. A.R.B., G.J., R.B., and A.Z. performed the analyses for the manuscript. A.R.B., G.J., A.Z., and R.B. wrote the manuscript with input from all authors.

## Competing interests

D.B.S. has served as a consultant for/received honoraria from Loxo Oncology, Lilly Oncology, Pfizer, QED Therapeutics, Vivideon Therapeutics and Illumina. M.E.A. received speaker and consulting fees from Invivoscribe, Biocartis, AstraZeneca, Bristol-Myers Squibb, Clinical Care Options, PVI Peerview, Janssen Global Services, LLC, Physicians' Education Resource, LLC. M.L. has received advisory board compensation from Boehringer Ingelheim, AstraZeneca, Bristol-Myers Squibb, Takeda, and Bayer, and research support from LOXO Oncology and Helsinn Healthcare. A.R.B. has stock ownership in Johnson & Johnson. S.R.Y has received consulting fees from Invitae. M.F.B. has received consulting fees from Roche and grant support from Illumina and Grail. M.F.B., D.T., P.S., J.P., B.H.L., M.H., and F.M. are co-inventors on a provisional patent application for systems and methods for detecting cancer via cfDNA screening. A.Z. received speaking fees from Illumina. B.H.L. receives royalties from BioLegend for development of products related to CITE-seq. J.H. has received research funding from Bayer, Eli Lilly, and Boehringer Ingelheim; and honoraria or consulting fees from Axiom Healthcare Stetegies, WebMD, Illumina, and Cor2Ed. K.N. has received honoraria from Biocartis. S.C. has received consulting fees from Sermonix, Paige.ai, Novartis, and Lilly. J.J.H. has received consulting fees from Bristol Myers Squibb, Exelexis, Eisai, Merck, ImVax, CytomX, and Eli Lilly and research support from Bristol Myers Squibb, the Wien Initiative in Liver Cancer Research, and the Society of MSKCC. H.Y. has consulted for AstraZeneca, Blueprint Medicine, Janssen Oncology and Daiichi. Her institution has received research funding for clinical trials from AstraZeneca, Daiichi, Pfizer, Novartis, Cullinan Oncology, Lilly. W.A. has received research funding from AstraZeneca, Zenith Epigenetics, Clovis Oncology, GlaxoSmithKline; and honoraria or consulting fees from CARET, Clovis Oncology, Janssen, MORE Health, ORIC Pharmaceuticals, Daiichi

Sankyo. L.M.S. is an employee of Loxo Oncology at Lilly. L.M.S. has consulted for Novartis, Pfizer, AstraZeneca and Roche Genentech; received research funding from AstraZeneca, Puma Biotechnology and Roche Genentech; travel or accommodations expenses from AstraZeneca, Pfizer, Puma Biotechnology and Roche Genentech; honoraria from Pfizer and AstraZeneca. D.T. has received research support from ThermoFisher Scientific, EPIC Sciences, speaking honoraria and travel support from Nanodigmbio, Cowen, BoA Merrill Lynch, D.T. and L.D. are co-inventors on a provisional patent application for systems and methods for distinguishing pathological mutations from clonal hematopoietic mutations in plasma cell-free DNA by fragment size analysis. R.B. has received a grant and travel credit from ArcherDx, honoraria for advisory board participation from Loxo oncology and speaking fees from Illumina. B.T.L. has served as an uncompensated advisor and consultant to Amgen, Genentech, Boehringer Ingelheim, Lilly, AstraZeneca, Daiichi Sankyo, and has received consulting fees from Guardant Health and Hengrui Therapeutics. He has received research grants to his institution from Amgen, Genentech, AstraZeneca, Daiichi Sankyo, Lilly, Illumina, GRAIL, Guardant Health, Hengrui Therapeutics, MORE Health and Bolt Biotherapeutics. He has received academic travel support from Resolution Bioscience, MORE Health, and Jiangsu Hengrui Medicine. He is an inventor on two institutional patents at MSK (US62/685,057, US62/514,661) and has intellectual property rights as a book author at Karger Publishers and Shanghai Jiao Tong University Press. B.T.L. is supported by the Memorial Sloan Kettering Cancer Center Support Grant P30 CA008748 from the National Institutes of Health. C.M.V. discloses relationships and financial interests with the following: DocDoc Pte. Ltd. (Provision of Services), Paige.AI, Inc (Ownership/Equity Interests; Provision of Services). E.O.: Research Funding to MSK: Genentech/Roche, Celgene/BMS, BioNTech, BioAtla, AstraZeneca, Arcus, Elicio, Parker Institute, AstraZeneca, Pertzye. Consulting Role: Cytomx Therapeutics (DSMB), Rafael Therapeutics (DSMB), Sobi, Silenseed, Tyme, Seagen, Molecular Templates, Boehringer Ingelheim, BioNTech, Ipsen, Polaris, Merck, IDEAYA, Cend, AstraZeneca, Noxxon, BioSapien, Bayer (spouse), Genentech-Roche (spouse), Celgene-BMS (spouse), Eisai (spouse). P.R. received institutional grant/funding from Grail, Illumina, Novartis, Epic Sciences, ArcherDx and Consultation/Ad board/Honoraria from Novartis, Foundation Medicine, AstraZeneca, Epic Sciences, Inivata, Natera, and Tempus. C.M.R. has consulted regarding oncology drug development with AbbVie, Amgen, Astra Zeneca, Epizyme, Genentech/Roche, Ipsen, Jazz, Lilly, and Syros. CMR serves on the scientific advisory boards of Bridge Medicines, Earli, and Harpoon Therapeutics. G.I. reports consulting/advisory role: Bayer, Janssen, Mirati Therapeutics, Basilea Pharmaceutical and Research Funding: Mirati Therapeutics, Novartis, Seagen, Inc, Bayer, Janssen, Debiopharm. Honoraria: The Lynx Group, DAVA Oncology. G.J., M.D., A.R., Y.H., M.H., J.S., B.J.M., T.B., I.J., R.P., A.B.R., I.R., A.A., H.W., D.P., D.N.B., A.S., X.J., E.G., J.L.Y., D.P.S., J.D., N.D., A.S., A.L., E.S.L., A.S.B., D.C.F., Y.L., D.A.M., R.S., T.B., J.K.B., J.C.C., S.D., M.H., C.M., D.S.R., V.E., J.Y., and B.K. have no relevant competing interests to declare.
