## [Peer Review File · Nature Communications]

Reviewer #1, expert in liquid biopsies (Remarks to the Author):

Thank you for addressing the initial review comments thoroughly. I have no further comments. I look forward to having this important work available.

Reviewer #3, expert in clonal evolution and therapy response (Remarks to the Author):

Brannon and colleagues describe a capture and UMI-based liquid biopsy assay called MSK-ACCESS which comprises a panel of MSK-IMPACT-informed 129 genes and allows the detection of cfDNA mutations at high sensitivity (92%) down to 0.5% VAF and specificity (99%). The assay is also able to filter both germline variants and mutations related with CH through a further control of WBC sequencing. Overall, this is a well-conducted study and well-written manuscript. The detailed comparison for the concordance between MSK-IMPACT and MSK-ACCESS assay on the some patients provides important information into the efficiency of liquid biopsy for guiding precision treatment of patients.

1. The address the reasons that can possibly explain the discordance between MSK-IMPACT and MSK-ACCESS assay on the same patients, the authors have investigated tumor content (purity), VAF, clonality and the time between sample collection (Δ DOP). Another important reason – whether the blood samples were collected prior to or after drug treatment (e.g. chemo- or targeted therapy) should also be addressed. Drug treatment is known to confer strong selection on the tumor clonal structure thus would have significant impact on the cfDNA mutation profiles. For example, the authors found dramatic increase of AR mutations by MSK-ACCESS as compared to MSK-IMPACT in prostate cancer (Fig. 2f), which is likely due to selection by treatment of hormone therapy for prostate cancer. Another examples are EGFR mutation in non-small cell lung cancer and NF1 mutation in breast cancer (Fig. 2f). The authors indeed found an association between the time between sample collection (Δ DOP) and mutational concordance between MSK-ACCESS and MSK-IMPACT. Is it actually explained by whether the treatment was given as longer Δ DOP might be related with higher chance of treatment?

2. The criteria for classifying clonal and subclonal mutations should be described in the methods.

3. It seems multiple blood samples were assayed by MSK-ACCESS for some patients (687 blood samples from 617 patients). Are these regarding multiple time points? How is the mutational dynamics along time?

4. Line 510, is “actionable mutations” a typo? My understanding is Fig. 3f shows all mutations while Supp. Fig 5 shows only actionable mutations.

5. In Lines 195-196, the authors describe “Seventy-three percent ($n = 498$) of all samples had at least one alteration detected”. However, in line 199, they said “we clinically reported a total of 1697 SNVs and indels in 486 samples”. Why there is inconsistency regarding the sample number, 498 vs 486?

Reviewers' comments:

Reviewer #1, expert in liquid biopsies (Remarks to the Author):

Thank you for addressing the initial review comments thoroughly. I have no further comments. I look forward to having this important work available.

Response: We thank the reviewer for their valuable comments and feedback that improved our communication of results in this manuscript.

Reviewer #3, expert in clonal evolution and therapy response (Remarks to the Author):

Brannon and colleagues describe a capture and UMI-based liquid biopsy assay called MSK-ACCESS which comprises a panel of MSK-IMPACT-informed 129 genes and allows the detection of cfDNA mutations at high sensitivity (92%) down to 0.5% VAF and specificity (99%). The assay is also able to filter both germline variants and mutations related with CH through a further control of WBC sequencing. Overall, this is a well-conducted study and well-written manuscript. The detailed comparison for the concordance between MSK-IMPACT and MSK-ACCESS assay on some patients provides important information into the efficiency of liquid biopsy for guiding precision treatment of patients.

1. The address the reasons that can possibly explain the discordance between MSK-IMPACT and MSK-ACCESS assay on the same patients, the authors have investigated tumor content (purity), VAF, clonality and the time between sample collection (Δ DOP). Another important reason – whether the blood samples were collected prior to or after drug treatment (e.g. chemo- or targeted therapy) should also be addressed. Drug treatment is known to confer strong selection on the tumor clonal structure thus would have significant impact on the cfDNA mutation profiles. For example, the authors found dramatic increase of AR mutations by MSK-ACCESS as compared to MSK-IMPACT in prostate cancer (Fig. 2f), which is likely due to selection by treatment of hormone therapy for prostate cancer. Another example are EGFR mutation in non-small cell lung cancer and NF1 mutation in breast cancer (Fig. 2f). The authors indeed found an association between the time between sample collection (Δ DOP) and mutational concordance between MSK-ACCESS and MSK-IMPACT. Is it actually explained by whether the treatment was given as longer Δ DOP might be related with higher chance of treatment?

Response: We thank the reviewer for their comment and we agree that any possible drug treatment between tissue biopsy and blood draw could also explain the discordances between MSK-IMPACT and MSK-ACCESS. As pointed out by the reviewer, the increase in AR mutations in prostate cancer by MSK-ACCESS reflects this. We have noted this difference on page 7 of the manuscript: “Overall, the alteration

rates in select genes and cancer types between MSK-ACCESS and MSK-IMPACT were comparable, with some notable exceptions such as such as a decrease in KRAS mutations in pancreatic cancer, an increase of EGFR mutations in lung, AR mutations in prostate cancer and NF1 mutations in Breast cancer in the ctDNA (**Figure 2F**)."

However, unfortunately, we do not have the treatment information for each patient across the samples we have utilized in this study as it is not readily available and is outside the scope of this manuscript. Future disease-specific manuscripts will shed a better light on the effect of oncologic therapy on the clonal changes into the tumor genomes and heterogeneity and how that is reflected in the plasma results. We have included this as a caveat in the discussion with the following text (page 12).

As comprehensive data on oncologic therapy becomes available, the detailed mechanisms of discordance between tissue and plasma derived mutations can be investigated further in future studies.

2. The criteria for classifying clonal and subclonal mutations should be described in the methods.

Response: We thank the reviewer for noticing the lack of a description. We added the following sentence to the methods:

A mutation was classified as clonal if the cancer cell fraction value obtained from FACETS was greater than 80%, otherwise the mutation was classified as sub-clonal.

3. It seems multiple blood samples were assayed by MSK-ACCESS for some patients (687 blood samples from 617 patients). Are these regarding multiple time points? How is the mutational dynamics along time?

Response: The reviewer is correct that for some patients we have sequenced multiple plasma samples. However, to make the analyses consistent, we have used only the first plasma sample for each patient in subsequent sections of the manuscript. We expect more disease focused manuscripts to tackle the question of mutational dynamics along time within the context of treatment modalities. Otherwise, we feel that clonal dynamics are outside the scope of the current manuscript.

4. Line 510, is "actionable mutations" a typo? My understanding is Fig. 3f shows all mutations while Supp. Fig 5 shows only actionable mutations.

Response: We thank the reviewer for pointing this out, indeed this was a typo and we have fixed it.

5. In Lines 195-196, the authors describe "Seventy-three percent (n = 498) of all samples had at least one alteration detected". However, in line 199, they said "we

clinically reported a total of 1697 SNVs and indels in 486 samples". Why there is inconsistency regarding the sample number, 498 vs 486?

Response: 498 samples harbored at least one of any kind of alteration (SNVs, indels, copy number alterations, or structural variant) while 486 samples specifically harbored at least one mutation (SNVs and indels). We have further clarified this in the text as follows:

Seventy-three percent (n = 498) of all samples had at least one alteration (mutation, copy number alteration or a structural variant) detected, with a non-zero median of 3 per patient (range 1-28)

Reviewer #3 (Remarks to the Author):

My comments have been well addressed and I recommend publication of this manuscript.